# Immediate Early Gene c-fos in the Brain: Focus on Glial Cells

**DOI:** 10.3390/brainsci12060687

**Published:** 2022-05-24

**Authors:** Fernando Cruz-Mendoza, Fernando Jauregui-Huerta, Adriana Aguilar-Delgadillo, Joaquín García-Estrada, Sonia Luquin

**Affiliations:** Neuroscience Department, Health Sciences Faculty, University of Guadalajara, Guadalajara 44340, Mexico; fernandocm649@gmail.com (F.C.-M.); fernando.jhuerta@academicos.udg.mx (F.J.-H.); adrianaaguilar1406@gmail.com (A.A.-D.); jgarciaestrada@gmail.com (J.G.-E.)

**Keywords:** c-fos, glial cell, astrocytes, microglia, oligodendrocytes, neurons

## Abstract

The *c-fos* gene was first described as a proto-oncogene responsible for the induction of bone tumors. A few decades ago, activation of the protein product c-fos was reported in the brain after seizures and other noxious stimuli. Since then, multiple studies have used c-fos as a brain activity marker. Although it has been attributed to neurons, growing evidence demonstrates that c-fos expression in the brain may also include glial cells. In this review, we collect data showing that glial cells also express this proto-oncogene. We present evidence demonstrating that at least astrocytes, oligodendrocytes, and microglia express this immediate early gene (IEG). Unlike neurons, whose expression changes used to be associated with depolarization, glial cells seem to express the c-fos proto-oncogene under the influence of proliferation, differentiation, growth, inflammation, repair, damage, plasticity, and other conditions. The collected evidence provides a complementary view of c-fos as an activity marker and urges the introduction of the glial cell perspective into brain activity studies. This glial cell view may provide additional information related to the brain microenvironment that is difficult to obtain from the isolated neuron paradigm. Thus, it is highly recommended that detection techniques are improved in order to better differentiate the phenotypes expressing c-fos in the brain and to elucidate the specific roles of c-fos expression in glial cells.

## 1. Introduction

The *c-fos* proto-oncogene was first described as the gene responsible for the induction of bone tumors by the Finkel–Biskis–Jinkins murine sarcoma virus [1]. It was observed in many species including humans, rats, mice, and chickens [2,3,4,5]. Its length was stated as around 4 kb organized in four exons that encode a unique mRNA and translate into a 55 kDa protein [6]. According to the GenBank of the National Center for Biotechnology Information (NCBI), the FOS family includes the members *Fos, Fos-B*, *Fra1* (FOSL1), and *Fra2* (FOSL2), which encode for proteins that contain a leucine zipper and can dimerize with the JUN family. Dimerization forms the activator protein 1 (AP1) complex, which works as a transcription factor (NCBI, 2020; Gene ID: 2353) (Figure 1).

The product of the *c-fos* gene is a protein named Fos that may act as a transactivator or transrepressor factor. For instance, dimerization with Jun exerts positive modulation, while binding with the activation transcription factor (ATF) leads to negative control. This activation versatility lies in their relative affinity for binding to various interacting complexes [7,8]. Cell-transforming properties of the fos gene are related to a c-terminus transactivator domain (c-TAD), which is present in Fos and FosB [9]. Since c-fos is not derived from the cell cycle progression and is quickly induced by extracellular stimuli [10,11], it has been identified as an immediate expression gene (IEG) [12].

Fos and *c-fos* genes have been associated with proliferation, differentiation, transformation, and cellular death [13,14,15,16]. Rapid and transiently, c-fos can be expressed by almost every cell as an important piece in cell signal transduction derived from growth, maturation, factor release, or physic stress (wounds, heat, etc.). It can be induced by many stimuli, including the protein kinase C (PKC), intracellular calcium release, steroid hormones, and neurotransmitters [6,17,18,19,20]. Since its expression is transitory in synchronized cells, it can be difficult to observe in non-synchronized cells [16]. Thus, c-fos is a transcription factor whose expression is quickly and synchronically induced; many studies have promoted its use as a marker of stimuli-induced changes in brain activity. Expression changes have mostly been attributed to neurons; however, as glial cells are the main regulators of growth, regeneration, and inflammation in the brain, it is highly expected that some of the c-fos-expressing cells correspond to glial phenotypes. Next, we will explore the main data linking c-fos to the brain, and then, we will focus on evidence specifically linking this marker to the glial phenotypes.

## 2. c-fos in the Brain

At the end of the past century, some IEGs were associated with stimuli–transcription activity in the brain. The most commonly used reporter for this phenomenon was the *c-fos* gene, whose expression in neurons was described to be transient [21]. As a marker of cellular activity, c-fos was subsequently used to identify brain regions implicated in stimuli processing [22,23] or relationships between two or more brain areas [4]. Similar to electrophysiology, it was used to measure neuronal activity but also provided information about morphology and cell type both in vivo [24,25,26,27] and in vitro [28,29]. The most commonly used methods to reveal c-fos changes became immunohistochemistry and Western blotting (in situ hybridization). In addition, c-fos expression was also employed in transgenic models (i.e., fos-LacZ) to inactivate some genes in a quantitative manner [10,30,31,32]. More recently, optogenetic models used c-fos to mark and manipulate specific brain nuclei [33].

Former studies investigating the expression of c-fos in neuroblastic cells supported the use of this marker to evidence changes in brain activity. Neuroblastic cultures demonstrated that nerve growth factor (NGF) [10], phosphatidic acid [34], epidermal growth factor (EGF), fibroblast growth factor (FGF), platelet-derived growth factor (PDGF), insulin, and potassium chloride (KCl) could induce the expression of c-fos in these cells [35]. Following experiments characterized the main cell pathways able to activate c-fos in neural cells [10,34,35]. It was demonstrated that growth factors, interferons, interleukins, calcium release, and G-protein ligands could initiate phosphorylation cascades, affecting c-fos expression. The phosphorylation proteins Jak 1-2, tyrosine kinase 2 (Tyk2), calmodulin kinases (CaMks), and protein kinase A (PKA) were described as key components of these pathways. The second messengers cAMP, GTP, and Ca^2+^ were evidenced as main mediators. Signaling complexes such as ras-raf, MEK-MAPK, or cAMP Response Element-Binding proteins (CREB) were also recognized in the c-fos signaling map. The list of transcription regulators affecting c-fos expression included the serum response element (SRE), cAMP response element (CRE), SIF-inducible element (SIE), interferon stimulation response element (ISRE), and IFN-γ-activated site (GAS) [6,11,36,37]. Pathways affecting c-fos expression are summarized in Figure 2.

To our knowledge, c-fos was initially used in brain research to measure the neural activity after seizures. In this area, it allowed researchers, for example, to map the neuronal pathways involved in different models/intensities of the seizure [38,39,40]; to estimate the effect of antiepileptic drugs [41]; or to analyze new-born cells and asymmetries after seizures [42,43]. c-fos has also been consolidated as a reliable marker for cell activation derived from learning [14] and memory [44,45]. The use of c-fos expression permitted researchers to predict cognitive worsening in Alzheimer’s disease and other conditions [46,47]. The relevance of sleep for memory acquisition and synaptic plasticity was also investigated by measuring c-fos during REM sleep [48]. Furthermore, c-fos was crucial to elucidating the role of the amygdala, thalamus, and hypothalamus in conditioned fear processing [49] and the epigenetic regulation of neuroplasticity [50], and cognition [51].

Studies of psychiatric disorders and their therapeutics also benefited from the use of c-fos. Fos, Fos-B, Jun, and Egr1 were reported to be upregulated in patients suffering from schizophrenia [52]. The antidepressant effect of optogenetic stimulation in the medial prefrontal cortex (mPFC) was assessed by mapping c-fos expression [53], and some antipsychotic drug effects in the brain were located through fos expression [54,55]. Stress studies, on the other hand, demonstrated activity changes in the frontal cortex and many limbic structures [56,57,58] that may predict susceptibility [59] or resilience to stress [60]. Moreover, it has been suggested that c-fos/Fos-B could be reliable markers for investigating the adaptive capabilities of the brain under stress conditions [61,62]. In addition, the study of endocrine responses demonstrated that suckling can modify the c-fos expression in the cerebral cortex [63], and that metabolic dysfunction produced by diabetes activates c-fos in the bed nucleus of stria terminalis (BNST) [64]. Research on circadian cycles and their alterations also evidenced c-fos changes in the suprachiasmatic nuclei (SCN), the main pacemaker in mammals [65,66]. Hypothalamic changes in c-fos levels may be used to measure altered circadian rhythmicity [67], intraspecific alternative chronotypes [68], or differences between diurnal and nocturnal animals [69,70].

Models inducing brain injury evidenced that c-fos increases 3 h after damage but also 3 days later. Thus, c-fos can accompany the immediate neuroprotective effects after brain edema, but also the delayed apoptosis in later stages of the lesion [71]. On the other hand, models inducing pain in the brain confirmed c-fos’s ability to map the brain areas involved in nociception [72,73], and allowed researchers to elucidate the role of periaqueductal gray and adenylate-cyclase-activating polypeptide-38 (PACAP-38) [74] in nociception [75]. In addition, fos immunoreactivity permitted researchers to describe a non-canonical auditory nociceptive system evidencing that cochlear nuclei could be active in deaf mice exposed to noxious levels of noise (120 dB) [76]. Thus, it is clear that c-fos has a crucial role in signal transduction across the brain, but studies have prioritized neuronal cells. As mentioned before, c-fos could potentially be expressed in every cell type, and that could include the non-neuronal residents of the brain, the glial cells.

## 3. Glial Cells

Although glial cells have been acknowledged for their supportive role in maintaining neurons and brain homeostasis for years, a number of studies have shown that these non-neuronal cells are active regulators for the majority of the brain functions. For instance, glial cells regulate immune and stress responses [77,78], formation, development, and pruning of synapses [79], neurotransmitter uptake [80], blood–brain barrier formation [81], axonal guidance [82], and myelinization [83]. Glial cells have been typically grouped into two great groups: macroglia and microglia, which in turn subdivide again to describe more specific cell phenotypes.

Macroglia include astrocytes, oligodendrocytes, NG2 cells (oligodendroglial precursor cells), and ependymal cells [78]. Macroglia possess many physiological roles, ranging from neural progenitors (i.e., neurogenesis, astrogenesis, oligodendrogenesis) to structural pieces of the brain (the classic supportive role of glia) [84].

Astrocytes are perhaps the most versatile cells in the brain. They are one of the integral components of the synapse (tripartite synapse) [85,86], play a major role in the uptake and metabolism of neurotransmitters [80,87], modulate the energetical metabolism [88,89], promote and support neuronal migration [90,91], maintain the ionic balance in the synaptic and extrasynaptic space [92,93], take part in the blood–brain barrier [81,94], and represent the main source of neurogenesis and gliogenesis [82,95,96]. Astrocytes can be classified as fibrous, protoplasmic, and radial astrocytes, mainly attending to their morphology and localization. Figure 3 illustrates this.

Oligodendrocytes are mainly known for their role as myelinating cells. They are responsible for the optimal conduction of neuronal impulses and are able to influence and modify electrical activity [83,97,98]; in addition, oligodendrocytes supply iron to the brain, provide growth factors, and offer metabolic support [99,100,101,102]. Brain myelinating cells can be classified as interfascicular, perivascular, or perineuronal [101,103].

NG2 cells are also known as oligodendrocyte precursor cells (OPCs). They are multipotent glial cells that can carry out morphological and proliferative changes in response to several stimuli, such as demyelination and injury [104,105]. OPCs can also induce proinflammatory responses through cytokine release [106] or through regulation of microglial transition into states of activation [107]. These cells can differentiate into other glial cell types [108] or even neuronal phenotypes [109,110].

Ependymocytes are the cells lining the ventricular walls and the ones that produce the cerebrospinal fluid [111]; they can also help and guide neuronal migration through cerebrospinal currents [112]. Ependymal cells are derived from radial glia, and even though their proliferative properties remain controversial, evidence suggests they have no proliferative role in the mature brain [113].

Microglia, on the other hand, were first described as the innate immune cell of the brain [114]. Beyond immune function, microglia also influence the pruning of dendrite processes, regulate the physiochemical microenvironment, influence the production and release of growth and survival factors, and promote apoptosis and cleaning of cell detritus [115,116,117]. Many of these roles depend on their activation state, which, according to morphological changes, can be classified as M0 (resting), M1 (pro-inflammatory, classical, or macrophage activation), and M2 (anti-inflammatory or alternative activation) [114,118]. Figure 4 illustrates this.

Hence, glial cells exert crucial roles for the correct functioning of the brain, ranging from proliferation to communication, both under physiological circumstances and pathological conditions. c-fos has been shown to be crucially involved in most of these conditions; so, it is not illogical to think that this IEG may be specifically involved in glial cell activity and signaling. On the contrary, as c-fos a rapidly activated gene in stimulated cells, it may also provide information on the glial activation patterns. Thus, in the next section, we will summarize the findings supporting this idea.

## 4. c-fos and Glial Cells: Evidence and Perspectives

As mentioned before, glial cells are the most abundant and versatile cells in the brain; they interact with neurons, blood vessels, and other glial cells to regulate the many functions of the brain. We stated that they act in coordination and interact with each other to maintain homeostasis and shape the distinctive actions of the CNS. We also summarized the current knowledge of c-fos dynamics and highlighted the crucial role of this family of genes in promoting and maintaining the homeostasis of the CNS microenvironment. Now, we propose that glial cells are sensitive to the actions of the IEG c-fos.

To our knowledge, the first set of studies exploring in vivo expression of c-fos in glial cells was published at the end of the 1980s, when researchers investigated the nature of proliferative cells in the surroundings of a damaged area. These studies suggested that some of the cells expressing c-fos near the injured area corresponded to a glial phenotype. Since the proliferation of neurons was not expected, initial observations assumed that c-fos-positive new-born cells were glial cells. Complementary experiments allowed researchers to demonstrate that at least some of the c-fos-expressing cells could be GFAP-positive (astrocytes) [119,120]. Since then, astrocytes have been the most investigated cell type regarding c-fos expression.

### 4.1. c-fos in Astrocytes

Initial studies investigating c-fos in this population explored implications in proliferation/maturation. Former in vitro experiments evaluated whether the mitogenic agents EGF, FGF, tetradecanoyl phorbol acetate (TPA), dbcAMP, or forskolin were able to induce c-fos in astrocytes. They found that these agents strongly induce c-fos and that major expression rates should be expected from 20 to 45 min after treatment [121]. In vitro stimulation of neurotransmitter systems also induced c-fos in astrocytes. Carbachol (cholinergic agonist), norepinephrine (NE), isoproterenol (ISO; β-adrenergic agonist), and phenylephrine (PHE; α-adrenergic agonist) were used to demonstrate that stimulation of c-fos through cholinergic or adrenergic pathways can modulate secondary genes or induce phenotypic changes [122]. The role of c-fos in astrocyte proliferation and differentiation was also explored by using mitogens (EGF, bFGF, db-cAMP, TPA) or depolarizing conditions (elevations in Ca^2+^ uptake or high concentrations of K^+^). Results showing that mitogens but not depolarization enhanced the expression indicate that c-fos could be specifically involved in astrocyte proliferation/differentiation [123]. Additionally, serotonin induces c-fos in astrocytes through its receptor 5HT_2B_R, which in turn enhances calcium release, metalloproteinases, and EGF release [20]. The use of endothelins to stimulate astrocytes corroborated that c-fos might be implicated in NGF expression during brain development [124]. Later in the 1990s, it was evidenced that the calcitonin-gene-related peptide (CGRP), a molecule produced by damage, was able to induce dose–response expression of c-fos in astrocytes. Since forskolin (an adenylate cyclase activator) reduced its effect, it was supposed cAMP had a role in the induction of c-fos associated with transformation and reparation of the injured brain [125].

Immune system mediators are also implicated in the glial expression of IEGs. c-fos and c-jun have been reported to be increased in astrocytes exposed to IFN-γ in a dose–response manner [11]. The IFN-γ-induced expression of c-fos in astrocytes regulates the complement factor H, whose abnormal levels instead induce neuronal loss in pathologies such as Alzheimer’s disease [126]. Other cytokines (TNF-α, IL-1β, and IFN-γ) and lipopolysaccharides (LPS) involved in inflammation modulate the expression of c-fos in astrocytes. LPS, LPS+ IL-1β, and IFN-γ induce c-fos in these cells. TNF-α, on the other hand, may enhance the LPS-induced increases in c-fos [127]. LPS induction of c-fos involves the p38 MAPK pathway, which activates Elk1, CREB/ATF-1, and later the SRE or CRE promoter [128]. Astrocytes infected with the adenovirus Ad.βGal expressed both c-fos and the apoptotic marker caspase-3, suggesting that c-fos can also indicate apoptosis [129]. Moreover, experimental autoimmune encephalitis allowed researchers to characterize a subpopulation of c-fos-expressing astrocytes named ieastrocytes. In this experimental model, the reporter system TetTag/green fluorescent protein was used to reveal the historical activity of c-fos in astrocytes. Promising results suggest astrocyte c-fos activity is a biomarker for autoimmune encephalitis [3].

Experimental models of damage also evidenced the activity of c-fos in astrocytes. In vitro models of heat shock and scratch wound showed not only that astrocytes express c-fos, but also that quercetin can inhibit the hypertrophy induced by scratches. That suggests that reactive astrogliosis could be associated with c-fos expression [130]. Experimental ocular hypertension models also demonstrated c-fos expression in astroglia. Monkeys with experimental glaucoma and astrocyte cultures of human glaucomatous optic nervous were found to overexpress c-fos [32]. Ischemia was also reported to induce quick and transient expression of the c-fos gene in cultured astrocytes. Those experiments exhibited that astrocytes rapidly increase the expression of c-fos after 30 min, reach a maximal expression level after 60 min, and diminish their expression levels after 2 h [131]. Chemical hypoxia, in turn, reverted the enhancing effect of ATP on the expression of c-fos [132]. Mimicking excitotoxicity, it was found that glutamate stimulation of astrocytes can rapidly increase the expression of c-fos since peak levels were reached 1 h after exposure [133]. Glutamate enhancements of c-fos could be mediated by mGluR5 and calcium dynamics since the addition of BAPTA (a calcium chelator) inhibits this enhancement [134].

Other conditions have been demonstrated to induce c-fos in astrocytes. Angiotensin II (AngII), for instance, can differentially regulate IEGs with lower increases in c-jun in contrast to c-fos. For some researchers, AngII dysregulation could lead to pathological responses through modulation of astrocytic c-fos expression [135]. The antidepressant fluoxetine was also evidenced to show a dual response varying with the effect of the dose; higher doses (5, 10 μM) enhance c-fos expression through ERK1/2, while lower doses (0.5, 1 μM) inhibit astrocyte expression of c-fos by the Akt pathway [136]. Conditions such as stress exposure or cognitive assessments have also evidenced c-fos induction in this population. It has been reported, for example, that learning activates c-fos in hippocampal astrocytes [137] and that restraint stress increases GFAP/c-fos+ in exposed subjects [138]. Table 1 summarizes these works.

### 4.2. c-fos in Oligodendrocytes

Although astrocytes represent the vast majority of experiments investigating c-fos in glial cells, oligodendrocytes have also been the target of some experiments. Like astrocytes, growth factors (bFGF, EGF, PDGF, and IGF-1) were reported to stimulate proliferation or maturation in OPCs, also known as NG2-glia. It was then reported that PKC and c-fos were required for this effect since blocking with H-7 (a PKC inactivator) resulted in inhibited proliferation [144,145]. The idea of c-fos as a promoter of proliferation/maturation of oligodendroglia became strengthened when experiments demonstrated that maturation of NG2 cells was preceded by c-fos expression [146]. Moreover, reports showed that c-fos-expressing oligodendrocytes exhibited a dose-dependent decrease in proliferation [147]. Oligodendrocyte proliferation can also be induced by carbachol, a cholinergic analogous that acts as a growth factor and also activates PKC and c-fos [148]. Norepinephrine, on the other hand, was also proved to exert a c-fos proliferative effect on oligodendrocytes. Calcium dependence of this process was suggested since sensitive G proteins, inositol phosphate 3 kinase (IP3K), and PKC were also activated [149].

Some pathologic conditions also affect the expression of c-fos in oligodendroglia. It was demonstrated that oligodendrocytes can express c-fos under glutamate stimulation too. Maximal expression levels were reported 60 min after exposure, and returns to basal levels were observed 6 *h* later. Glutamate experiments demonstrated that c-fos induction is mediated by AMPA-R and KA-R but not NMDA-R since specific antagonist CNQX and DNQX inhibited the effect, but MK-801 (NMDA-specific antagonist) failed to inhibit c-fos expression. Hypoxic conditions are also able to induce c-fos in oligodendrocytes. Hypoxia models demonstrated that oligodendrocytes express c-fos as an event preceding myelin loss, axonal damage, and apoptotic death [150]. There is also evidence that c-fos expression is stimulated by the hallucinogen d-LSD in vivo, an action possibly linked to the modulation of neuronal impulses or growth factor production [151]. Finally, it was reported that c-fos expression diminished with maturation in oligodendrocytes, but ethanol consumption retarded this decrease as well as the myelin basic protein (MBP) production [152]. These works are summarized in Table 2.

### 4.3. c-fos in Microglia

Given the mesodermal immune origin of microglia, it is highly expected that c-fos modulates some actions in this lineage. Even so, less research has explored this, with some pieces of evidence indicating that c-fos can be expressed by the immune residents of the brain. For instance, there are experiments showing that excitotoxic glutamate stimulation, through any of its ionotropic receptors or group I of metabotropic receptors, is able to induce c-fos expression and microglial activation [153]. In addition, it was reported that stimulation with kainic acid (KA) increases the expression of MHCII and class II transactivator (CIITA) in microglia, but coincidently, both are inhibited by pretreatment with triptolide, which actually decreases the phosphorylation of c-fos and c-jun and the consequent formation of AP1 [154].

Proinflammatory effects have been proposed for the expression of c-fos in microglia. It is known that microglial NOD-like receptor 3 (NLRP3) is associated with neuroinflammation and is also a therapeutic target in Alzheimer’s disease. Thus, the anti-inflammatory effects of dexmedetomidine inhibit NLRP3-derived inflammasome, modulating the c-fos upregulated expression [155]. As in other glial types, there is also evidence showing that LPS can induce the expression of c-fos in microglia and the consequent inflammation in the brain [156]. Additionally, studies of paraquat, an herbicide that is associated with a higher incidence of Parkinson’s disease, have evidenced that this substance can induce c-fos expression, as well as HSP60 and TLR4, which then increases the proinflammatory cytokine production and accelerates inflammatory responses [157]. We recapitulate these studies in Table 3.

## 5. Conclusions and Perspectives

Here, we compiled evidence supporting the relevance of c-fos for brain functions. Beyond neurons whose implications have been well-documented, we showed that glial cells represent an unexplored but very attractive target for c-fos investigation. Beyond depolarization, whose actions have been well-documented in neurons, the available evidence supports that c-fos expression may also accompany proliferation, maturation, repair, and damage in the brain. All of these functions are intensely linked to glial functions.

As stated above, astrocytes have been by far the most attended cells regarding c-fos expression in glial cells. It seems that proliferation/reparation are the phenomena most strongly linked to c-fos expression in this glial type. Thus, identification of c-fos changes in this phenotype could be useful to rapid and reliably predict physiological or pathological responses to specific stimuli. Reactive gliosis, scar formation, and other astrocyte actions involving proliferation/transformation could be predicted by measuring c-fos dynamics.

Oligodendrocyte research permitted researchers to posit that this activity marker may also be involved in differentiation, maturation, and obviously myelin formation. In this case, c-fos expression could be helpful in gaining knowledge on how the brain responds to stimuli affecting axonal conductivity. It is worth mentioning the relevance of the NG2 cells not only as oligodendrocyte precursors but also as specialized/differentiated cells able to support and contribute to the good functioning of the brain. Thus, exploration of c-fos changes in this subpopulation is now mandatory.

Microglia, on the other hand, evidenced that c-fos may be a powerful tool with which the immune/inflammatory responses conducted in the brain can be elucidated. Given that many cytokine genes are regulated cooperatively by a transcription factor complex that includes AP-1, c-fos could be an accurate sensor of inflammation in the brain and a reliable marker for the progression of some inflammatory diseases.

As studies investigating the link between c-fos and glial cells are still scarce, and many questions remain unanswered, we can now suggest the need for an effort to increase the specificity to evaluate neuronal and non-neuronal activity through c-fos.

First, we would like to conclude by affirming that the evidence presented here allows researchers to maintain that studies evaluating brain activity through c-fos must consider the possibility that some of the activated cells could be glial cells. Second, we would like to invite the scientific community to attend the question of the role of c-fos in glial cells. Answers on how, where, or when specific cells express c-fos will strongly contribute to the understanding of the brain function. We would also like to urge the community to determine and use adequate approaches to accurately define phenotypes involved in the cellular response to a myriad of physiological and/or pathological stimuli commonly assessed by c-fos expression. By doing this, we can make sure that brain activity changes correspond to different cell phenotypes and gain knowledge on how glial cells participate in the many functions of the brain.

## Figures and Tables

**Figure 1 brainsci-12-00687-f001:**
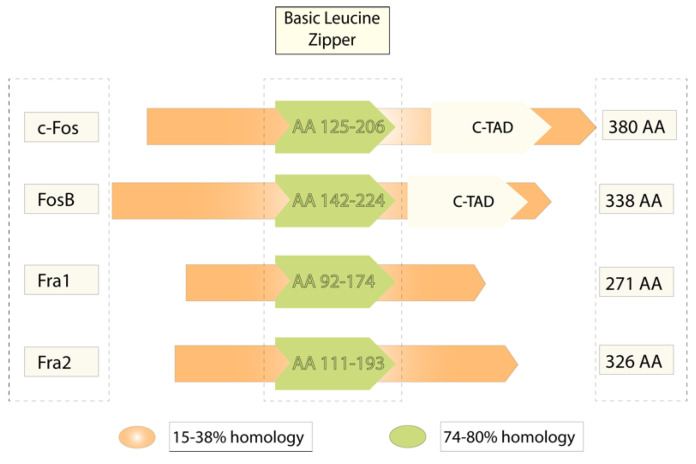
The Fos gene family members. The picture shows a comparative structure of the products of the fos gene, all of which contain a basic leucine zipper domain, but only c-fos and FosB have a c-terminal transactivator domain (c-TAD). It also indicates the high homology level in the basic leucine zipper domain, which is more variable in other regions.

**Figure 2 brainsci-12-00687-f002:**
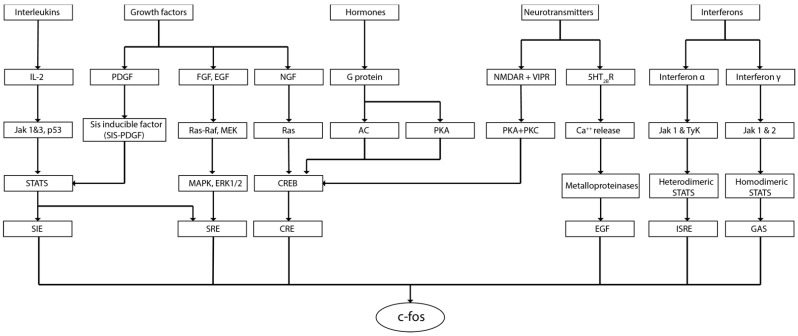
Cell signaling pathways that might induce c-fos expression.

**Figure 3 brainsci-12-00687-f003:**
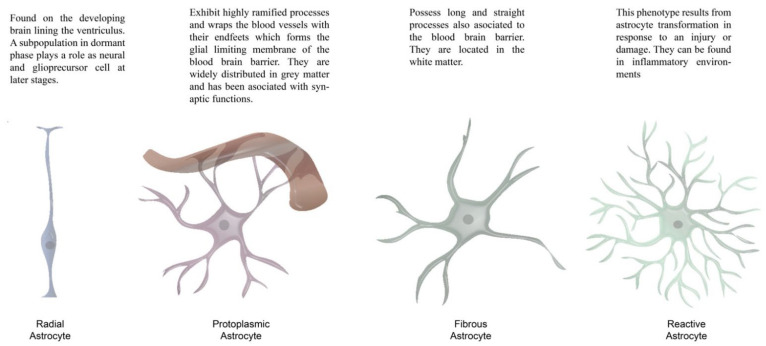
Illustration of the main astrocyte subtypes.

**Figure 4 brainsci-12-00687-f004:**
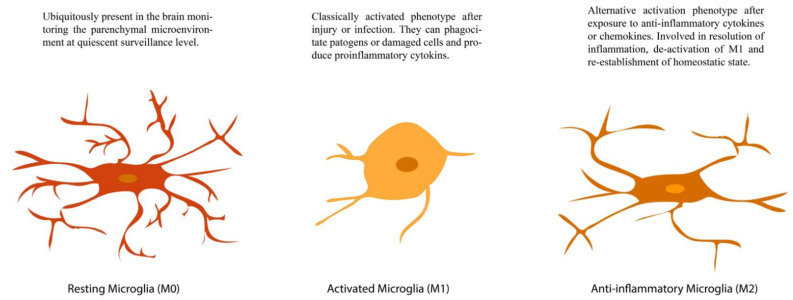
Activation states in microglial cells. Schematic representation of an M0 (**left**), M1 (**center**), and M2 (**right**) microglia.

**Table 1 brainsci-12-00687-t001:** Papers evidencing c-fos expression in astrocytes. Arrows indicate ↑ increased expression or ↓ diminished expression under different stimulation models.

Paper	Effect	Model	Approach	Methodology	Studied Area	Species
[119]	↑ After 1 h	Heat shock insult	In vivo	Immunohistochemical	Thalamus, hippocampus, corpus callosum, internal capsule, and fornix/fimbria	Rat
[121]	↑ After 30 min	Mitogens and growth factor exposure	In vitro	Northern blot	Primary cultures of cortical astrocytes (In secondary cultures)	Rat
[122]	↑ After 30–60 min	Muscarinic and adrenergic agonist exposure	In vitro	Northern blot	Primary cultures of cortical astrocytes (In secondary cultures)	Rat
[123]	-↑ After 30 min mRNA and 2 h protein -No change	-Mitogen exposure -Depolarizing conditions	In vitro	Northern blot and immunohistochemical	Primary cultures of neocortical astrocytes	Rat
[125]	↑After 30 min	Damage-associated molecular pattern (DAMPs)	In vitro	Northern blot	Primary cultures of cortical astrocytes	Rat
[124]	↑After 30 min	Endothelin exposure	In vitro	Northern blot	Rat astrocytoma C6 cells (C6-S and C6-V subclones) and primary cultures of cortical and striatal astrocytes	Rat/Mouse
[139]	↑After 30–60 min	Scratch wound of culture astrocytes	In vitro	Quantitative reverse transcriptase polymerase chain reaction (RT-PCR	Primary cultures of cortical astrocytes	Rat
[131]	↑0.5–2 h, peak at 1 h	Ischemic model (mineral oil)	In vitro	RT-PCR	Primary cultures of cortical astrocytes	Rat
[11]	↑After 30 min	Proinflammatory factor exposure	In vitro	Northern blot and flow cytometry	Primary cultures of cortical astrocytes	Mouse
[132]	↑After 15–60 min	Chemical hypoxia (0.5 mM cyanide for 1 h)	In vitro	Northern blot	RBA-2 type 2 astrocytes cell line	Rat
[130]	↑ NA	Heat shock insult	In vitro	Western blot	Primary cultures of astrocytes	Mouse
[140]	↑1 h, peak at 2 h	LPS administration	In vivo	Immunohistochemical	Hypothalamic supraoptic nucleus, posterior and anterior pituitary	Rat
[129]	↑0.5–1 h, peak at 30 min	Adenovirus (Ad.βGal) exposure	In vitro	Northern blot	Primary cultures of cortical astrocytes	Mouse
[127]	↑1 h	Proinflammatory factor exposure	In vitro	Northern blot	Primary cultures of astrocytes	Rat
[134]	↑After 15–30 min, peak at 30 min	Glutamate stimulation in excitotoxic levels	In vitro	Northern blot and immunohistochemical	Primary cortical glial cell cultures	Rat
[141]	↑After 1 h	Bradykinin exposure	In vitro	Western blot and RT-PCR	RBA-1 cell line	Rat
[126]	c-fos binding to mCFH promoter	NA	In vitro	Electrophoretic mobility shift assay (EMSA) and supershift assay	Astrocyte 2.1 (Ast 2.1) cell line and primary astrocytes, microglia and oligodendrocytes cultures	Mouse
[142]	Nuclear translocation	Amitriptyline exposure	In vitro	Western blot and real-time PCR	Primary cultures of astrocytes	Rat
[143]	↑After 1 h	Forskolin and IL-1 exposure	In vitro/ in vivo	Western blot and qPCR	Human cortical astrocytes/KO mouse	Human/ Mouse
[136]	↓ After 1 h at low doses (0.5–1 μM) ↑After 1 h at high doses (5–10 μM)	Fluoxetine exposure	In vitro	Western blot and RT-PCR	Primary cultures of astrocytes	Mouse
[137]	↑ After 90 min	Viral vector injection and CNO administration	In vivo	Immunohistochemical	Hippocampus	Mouse
[3]	↑ NA	Experimental autoimmune encephalomyelitis (EAE)	In vivo	Immunolabeling-enabled three-dimensional imaging of solvent-cleared organs (iDISCO) and flow cytometry	TetTag-cFos reporter mice	Mouse

**Table 2 brainsci-12-00687-t002:** Papers evidencing c-fos expression in oligodendrocytes.

Paper	Effect	Model	Approach	Methodology	Studied Area	Species
[144]	↑After 30 min	Mitogenic and growth factors exposure	In vitro	Northern blot and immunohistochemical	OPCs isolated from mixed glial cell cultures	Rat
[145]	↑After 0.25–8 h, peak at 1 h	Basic fibroblast growth factor (bFGF) exposure	In vitro	Northern blot and immunohistochemical	OPCs isolated from mixed glial cell cultures	Rat
[147]	↑After 0.25–6 h, peak at 1 h	Glutamate exposure	In vitro	Northern blot	OPCs isolated from mixed glial cell cultures	Rat
[148]	↑After 30–60 min	Carbachol exposure	In vitro	Northern blot and immunohistochemical	OPCs isolated from mixed glial cell cultures	Rat
[149]	↑After 30–60 min	NE exposure	In vitro	Western and Northern blot	OPCs isolated from mixed glial cell cultures	Rat
[150]	↑ 1, 6, and 9 days after induction with progressive increases	Experimental anterior optic nerve ischemia	In vivo	Quantitative real-time PCR (qRT-PCR) and immunohistochemical	Optic nerve	Mouse
[151]	↑After 90 min	d-LSD exposure	In vivo	Immunohistochemical	Prefrontal cortex	Rat
[152]	Delayed downregulation of c-fos levels during differentiation	Ethanol administration	In vitro	Western blot	CG-4 glial cell line	Rat
[146]	↑After 2 hours/day of SMF stimulation (0.3 T) for a period of 14 days	Static magnetic field (SMF) stimulation	In vitro	qRT-PCR	Human OPCs derived from induced pluripotent stem cells	Human

**Table 3 brainsci-12-00687-t003:** Papers evidencing c-fos expression in microglia.

Paper	Effect	Model	Approach	Methodology	Studied Area	Species
[153]	↑After 30 min (mRNA) ↑After 2 h postexposure (protein)	10-50-fold higher doses of glutamate than physiological exposure	In vitro	RT-PCR, Western blot, and immunohistochemical	Primary cortical microglial cells	Rat
[154]	↑p-c-fos after 2 h (KA)	Kainic acid (KA) exposure	In vitro	Western blot	BV-2 microglia cell line	Mouse
[155]	↓ Dose-dependent (Dex)	LPS/ATP and Dexmedetomi-dine (Dex) treatment	In vitro	PCR, Western blot, and immunohistochemical	Human microglia clone 3 cell line (HMC3)	Human
[156]	↑2 h after treatment (BV-2 cells) ↑2–6 h after treatment (hypothalamus)	LPS treatment	In vivo/in vitro	PCR, Western blot, and immunohistochemical	BV-2 microglia cell line/hippocampus and hypothalamus	Mouse/rat
[157]	↑6, 12, and 24 h of treatment	Paraquat and LPS exposure	In vitro	qRT-PCR	BV-2 microglia cell line	Mouse

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
