# Peer review of "Immediate Early Gene c-fos in the Brain: Focus on Glial Cells"

_brainsci, 2022, doi:10.3390/brainsci12060687_

Round 1

Reviewer 1 Report

Original Article Review
Journal: Brain Sciences

Title: Immediate early gene c-fos in the brain: focus in glial cells

Summary:

This is a review of the literature as it pertains to cFos and its expression in neurons and glial cells. After reading the title, I was very interested to read this paper. Immediate early gene expression in non-neuronal cell types is a topic that needs to be addressed. Overall, I felt that the authors provided a solid background on cFos expression followed by examples of it in each glial cell types and neurons. While I believe this review will make a great contribution to the field, there were a few aspects of the paper that could use improvement.

General comments:

  1. Abbreviations should be defined at the first mention of a term in the paper. There were a few instances throughout the paper in which abbreviation definitions occurred after its initial use, were never defined, or were not used correctly. Additionally, once something has been defined be consistent and use that throughout the paper.
  2. Throughout the paper, there were quite a few grammatical errors. Some things to look out for are verb tense and an improper or lack of article usage. Also be careful when using the phrase nonetheless.
  3. There was a lack of cohesion in section 2: c-Fos in the brain. Overall it felt more like a list of citations describing cFos expression in the brain without thought to the order they were included.
  4. Section 3:
    1. The figures could be better incorporated into the text. While the figures are visually appealing showing the different cellular states for microglia and astrocytes, this section could benefit from providing details on when and/or where these different glial cells are observed in the brain.
    2. The figure numbers in this section are not correct (both labeled as figure 4 and numbering in text is off as well).
    3. Adding subheadings to differentiate between the different glial cell types in the same order as those references in section four would provide more guidance to the reader and be consistent with the following section.
  5. Section 4:
    1. Table 1 formatting is off. Also, in the subsection 4.3, the text mentions that Table 1 summarizes studies in all glial cells, but in 4.1 it is specific to astrocytes. Please clarify.
    2. Is there any available information about which type of astrocyte subtypes or microglial activation states that express cFos? If so, please include this information in the relevant subsection.
    3. What does cFos expression in these glial cells mean? Is it a marker of glial cell activation as is the case with neurons? Does it differ by glial cell subtype (such as microglia=activation while astrocyte=proliferation)?

Reviewer 2 Report

This is good study.

Author Response

We thank the reviewer 2 for recognizing the quality of our manuscript.

Reviewer 3 Report

This is an interesting review of roles of the c-fos proto-oncogene in glia. The review provides a fairly comprehensive citation list that would allow a reader interested in the subject to gain insight into the current state of understanding. While the review raises many observations, there are a number of issues which could strengthen it:

Major point 1: The paper does not really explain why studying c-fos expression in glia is of interest. There are many genes expressed in glia- why focus on c-fos? What insight could studying this gene bring that other studies cannot? 

Major point 2: Can the authors speculate more in the discussion about possible roles of c-fos in glia? What could the gene be doing there? Why is that interesting?

Other points:

1. The manuscript needs to be carefully edited to make sure that grammar and word usage are appropriate. Reading is made quite difficult in spots because of such issues, and this distracts from the content.

2. Lines 55-57, the authors say that the first studies of c-fos in the brain were of rat pheochromocytoma cells. But these are adrenal cells (kidney). Please rephrase.

3. Line 77, explain the term Immediate Early Genes.

4. The statements about c-fos and activity are confusing. Neurons are constantly active throughout the brain. Yet, not all active neurons express c-fos. Please be precise in the statements connecting c-fos and activity.

5. Line 177, the phrase "Macroglia possess many physiological roles that came from neural progenitors to structural pieces of the brain [89]." is unclear. What is meant by this?

Author Response

We thank the reviewer 3 for his/her generous comments.

Round 2

Reviewer 3 Report

The authors have addressed the issues I raised with text changes. The manuscript is significantly improved in terms of scientific accuracy, and moderately improved in terms of English usage. I think the motivation for discussing fos should not be relegated only to the discussion- it should appear front and center in the abstract and introduction. As a reader, I want to understand from the outset why this review will be important to read.

Author Response

We thank reviewer 3 for his/her generous comments. The new version of the manuscript corrects some missed terms and introduces the c-fos - glia interaction in the abstract and introduction section.
